# COVID-19 Vaccination and Breastfeeding Mothers in Kahta District, Turkey

**DOI:** 10.3390/vaccines11040813

**Published:** 2023-04-07

**Authors:** Mehmet Emin Parlak, Erdoğan Öz, Osman Küçükkelepçe

**Affiliations:** 1Adıyaman University Training and Research Hospital, Adıyaman 02100, Turkey; meparlak02@harran.edu.tr; 2Adıyaman Province Health Directorate, Adıyaman 02100, Turkey; erdogan.oz@saglik.gov.tr

**Keywords:** breastfeeding, vaccine hesitancy, COVID-19, vaccine, breast milk

## Abstract

We aimed to determine the attitudes and behaviors of breastfeeding mothers regarding the vaccine by examining their knowledge of the COVID-19 virus vaccine and their hesitations about it. The research is a cross-sectional and descriptive study conducted in the Kahta district of Adıyaman, a southeastern province in Turkey, between January and May 2022. The study population consisted of 405 mothers who applied to the Kahta State Hospital Pediatrics outpatient clinic. A questionnaire form was used as a data collection tool, and a consent form was obtained from the participants. The vaccination rate (89%) of those who graduated from high school and above was significantly higher than that of those who graduated from secondary school or below (77.7%). As the economic situation worsened, the vaccination rate decreased. The vaccination rate (85.7%) of mothers whose breastfed child was 0–6 months old was found to be significantly higher than that of those with 7–24-month-olds (76.4%) (p:0.02). The rate of being vaccinated (73.3%) of those who had a new type of COVID-19 virus infection was significantly lower than the rate of being vaccinated (86.3%) of those who did not have a COVID-19 virus infection. The vaccination rate of those who received information from their family doctor and the internet was higher than that of those who received information from radio/TV and people around. The rate of mothers thinking babies should stop breastfeeding who graduated from secondary school or below was higher (53.2%) than the rate of mothers who graduated from high school or above (30.2%) to be vaccinated against the COVID-19 virus. To eliminate the hesitancy about vaccination in mothers, it is necessary to inform and educate the whole society correctly, starting with families with low education and economic levels.

## 1. Introduction

Thanks to its unique content, breast milk meets all of the baby’s nutritional needs for the first six months, half of it between six months and one year, and 1/3 of it between one and two years. Various substances in the content of breast milk not only protect the baby from diseases and protect the mother against some cancers. Breastfeeding positively affects the mother’s mental health and the child’s neuropsychological development due to the contact between mother and baby. However, 2/3 of babies worldwide cannot be fed with only breast milk for the first six months [1].

It is recommended that the baby be breastfed by its mother in the first hour after birth, including pandemic periods, exclusive breastfeeding for the first six months, and continued breastfeeding for up to two years without interruption [2].

According to the Turkey 2018 Demographic and Health Survey, 98% of children born in Turkey are breastfed for a certain period after birth. However, for various reasons, the rate of exclusive breastfeeding in the first six months remains at 41% [3]. Suspension or cessation of breastfeeding before 1–2 years adversely affects infant and maternal health [4].

The new type of coronavirus infection emerged in China in December 2019, and the first case in Turkey was detected in March 2020. The new type of coronavirus infection, which causes severe disruptions in all aspects of the world, especially in health, economy, education, and transportation, is milder in children. Only 0.2% of deaths worldwide are children under 5 [5].

During the COVID-19 pandemic, although the World Health Organization recommends that mothers who are infected with the COVID-19 virus or who have received the COVID-19 virus vaccine should continue to breastfeed their babies, in some countries such as Singapore, in line with the directives of health officials and health professionals, mothers who have or are suspected of having a COVID-19 virus infection are separated from their babies and continued feeding with breast milk is paused [6,7].

The inability to provide adequate education on the importance of breast milk, not only due to the COVID-19 virus infection but also due to the curfew, has led mothers who have had the COVID-19 virus vaccine to stop breastfeeding their babies or interrupt breastfeeding, especially mothers with low education and socioeconomic levels [8]. In addition, the unwillingness of mothers to regularly take their babies to health facilities due to the fear of infection caused the lack of adequate information by health professionals about the new type of COVID-19 virus disease, COVID-19 virus vaccine, and breastfeeding [9].

Despite tremendous vaccine progress, anti-vaccine opposition remains a severe threat to public health. Even if it is not in the form of opposition to vaccination, hesitancy to vaccinate increases the risk of preventable diseases turning into epidemics [10].

Anti-vaccination and vaccine hesitancy may cause a decrease in the vaccination rates of pregnant women as well as breastfeeding mothers. Increasing hesitation and concern about vaccines, even against conventional vaccines, maybe even more pronounced against the newly developed COVID-19 virus vaccine. So, in this study, we aimed to determine the attitudes and behaviors of breastfeeding mothers regarding the vaccine by examining their knowledge of the COVID-19 virus vaccine and their hesitations about it.

## 2. Materials and Methods

The research is a cross-sectional and descriptive study. The analysis was carried out in the Kahta district of Adıyaman, a southeastern province in Turkey, between January and May 2022. The study universe consisted of mothers who applied to the Kahta State Hospital outpatient clinic. It was planned to include at least 384 mothers with a 95% confidence level and a margin of error of 0.05. An offer was made to 421 mothers for the study, and 16 refused to participate. So, 405 mothers were included in the study, with a 3.8% non-acceptance rate. A questionnaire form prepared by the researchers was used as a data collection tool, and a written consent form was obtained from the participants.

Before starting the study, ethical approval from the non-interventional clinical research ethics committee of the relevant university and written permission from the institutions which would be involved in the study were obtained. The principles of the Declaration of Helsinki were conducted through study.

Analyzes were evaluated in 22 package programs of SPSS (Statistical Package for Social Sciences; SPSS Inc., Chicago, IL, USA). In the study, descriptive data are shown as n and % values in categorical data and mean ± standard deviation (Mean ± SD) values in continuous data. Chi-square analysis (Pearson chi-square) was used to compare categorical variables between groups. Bonferroni corrected p values were calculated to find the group from which the difference originates in the chi-square analysis. The statistical significance level in the study was accepted as *p* < 0.05.

All women entering the clinic were enrolled. Since it is the only hospital in the district, all people come to Kahta State Hospital to be examined. Therefore, the proportion of education level, economic status, internet, and radio/TV use reflected the general population.

The limitation of the study is that the study was conducted in a district hospital, and the mothers were selected only among those who applied to the hospital. Many mothers do not come to the hospital unless they have a severe problem. People who apply to the hospital come because they have a critical illness or a high level of consciousness. There-fore, the individuals included in the study may only partially represent part of the population.

## 3. Results

A total of 405 breastfeeding mothers with a mean age of 27.7 ± 4.3 (min = 19–max = 46) were included in the study. While the education level of 57.5% of the mothers was secondary school or below, 31.6% of them described their economic status as good. Of the participants, 19.3% of mothers had one child, 40.7% had two children, and 40% had three or more children. The children that 65.4% of the mothers breastfeed were 0–6 months old, and 34.6% were 7–24 months old. While 29.6% of the mothers had a new type of COVID-19 virus infection, 85.7% of their relatives had a new type of infection. Of the mothers, 82.5% were vaccinated against the COVID-19 virus. Of the mothers, 44% obtained information about COVID-19 from their family doctor, 17.5% from radio/TV, 21.2% from the internet, and 17.3% from people around them (Table 1).

While 68.4% of the mothers stated that it is safe for breastfeeding mothers to have the COVID-19 virus vaccine, 22.7% said they would breastfeed their baby 3 h after receiving the COVID-19 virus vaccine. While 39.5% of the participants stated that breastfeeding mothers should be vaccinated, 33.3% indicated that it would not upset them that they could not be vaccinated against the COVID-19 virus because they are breastfeeding. While 33.3% of the mothers think that the COVID-19 virus vaccination of the breastfeeding mother will also protect their baby, 38.8% believe that the baby should be expected to stop breastfeeding to be vaccinated against the COVID-19 virus. Of the mothers, 31.9% stated that they could not obtain precise information about the vaccination of breastfeeding mothers, and 44% of the health workers said they could not obtain a clear statement from the family physician (Table 2).

The rate of vaccination (89%) of those who graduated from high school and above was found to be significantly higher than the rate of those who graduated from secondary school or below (77.7%) (*p* = 0.003). There was a significant difference between economic status and vaccination status (*p* = 0.008). This difference was due to the difference between all three groups, and as the economic situation worsened, the vaccination rate decreased. The rate of vaccination (85.7%) of mothers whose breastfed child was 0–6 months old was found to be significantly higher than that of 7–24-month-olds (76.4%) (*p* = 0.020). The rate of being vaccinated (73.3%) of those who had a new type of COVID-19 virus infection was found to be significantly lower than that of those who had not had a new type of COVID-19 virus infection (86.3%) (*p* = 0.002). There was a significant difference between the COVID-19 virus vaccine information sources regarding vaccination status (*p* = 0.035). This difference is due to the difference between family physicians and those who use the internet and those who use radio/TV and people around. The vaccination rate among family physicians and those using the internet was higher than among those who benefited from radio/TV and people in the surrounding area (Table 3).

The rate of mothers who graduated from secondary school or below was found to be significantly higher (53.2%) than the rate of mothers who graduated from high school or above (30.2%) to receive the COVID-19 virus vaccine (*p* < 0.001). The rate (51.7%) of mothers who had a new type of COVID-19 virus infection that the baby should stop breastfeeding to be vaccinated for the COVID-19 virus was found to be significantly higher than the rate of mothers who did not have a new type of COVID-19 virus infection (40%) (*p* = 0.031). The rate of mothers who have been vaccinated with COVID-19 virus think that the baby should stop breast milk to be vaccinated with COVID-19 virus (40.1%) was found to be significantly lower than the rate of mothers who were not vaccinated with COVID-19 virus (59.2%) (*p* = 0.003) (Table 4).

There was a significant difference between educational status and information source (*p* < 0.001). This difference was due to the difference between the choice of family doctor and Radio/TV. According to this, the preference rate for a family physician was higher, and the preference rate for radio/TV was lower in those with high school or higher education levels than those with secondary education levels and below. There was a significant difference in information sources among the status of having COVID-19 (*p* < 0.001). This difference was due to the difference between choosing the family doctor, the Radio/TV, and the people around. In this study, patients with COVID-19 had a higher preference for family physicians and a lower rate of preference for radio/TV and people around them compared to those who did not have COVID-19 (Table 5).

## 4. Discussion

Feeding babies with breast milk until they reach the age of 2 is necessary for the physical and mental well-being of the mother and the baby. However, today, worldwide, breastfeeding is started late, breastfeeding rates and breastfeeding durations are gradually decreasing, and breastfeeding is interrupted for various reasons [11].

The most frequent reasons for decreased breastfeeding are insufficient knowledge about breastfeeding, sociocultural differences towards breastfeeding, educational factors of the societies, low socioeconomic level, mother’s and baby’s diseases, mother’s drug use, and vaccination [12,13].

During pandemic periods, breast milk should not be interrupted or discontinued due to the protection of children against infections [14]. In a study in which 23 studies were examined, it was found that the COVID-19 virus vaccine is safe for both pregnant and lactating mothers in terms of maternal and infant health; It has been reported that the side effects of the vaccine are similar to those of nonpregnant or non-breastfeeding mothers [15].

Pregnant and breastfeeding mothers are at risk of contracting the COVID-19 more severely. For this reason, it is essential to vaccinate both groups. However, vaccine hesitancy exists for all vaccines, including the COVID-19 virus vaccine, and the rate of hesitation is higher, especially during pregnancy and in lactating mothers [16,17].

Over time, COVID-19 vaccine hesitancy has subsided as more people are vaccinated. However, vaccination rates decreased again, as the continuation of the pandemic despite vaccination practices was thought to be more ineffective than the side effects of the vaccine [18].

A meta-analysis of 46 studies found vaccination hesitancy in lactating mothers was 53.1%, even higher than in pregnant women (49.2%). There are many factors in vaccine hesitancy; these are mainly: lack of sufficient information about the vaccine, the concern of mothers that the vaccine may cause infertility or other harm to their babies and themselves in the long term, the policies of the pharmaceutical companies regarding vaccines, the fact that the vaccine is produced in a short time, so side effects studies have not been conducted. Concern that the vaccine may have adverse effects on those with chronic diseases, that the vaccine may cause allergies, and the negative and often false posts of anti-vaccine people on social media and news platforms can be counted as the disincentive effect of the negative posts of family members and their friends about the vaccine [17].

In a study conducted at Ankara City Hospital in Turkey, the vaccination acceptance rate of mothers in the puerperal period was found to be 33.3%, 88.1% of whom were breastfeeding [18]. This rate was 39.5% in the present study.

In a survey conducted in America, 55.2% of lactating mothers said that they thought the COVID-19 virus vaccine could be safely administered. In contrast, for breastfeeding, 49.2% planned to be vaccinated, and 3.3% stated that they had already been vaccinated. 22.7% of the lactating mothers surveyed were undecided about whether to accept the vaccine. Those who refused the vaccine were worried that the vaccine would harm them or their baby. Some mothers thought the vaccine had a sterile effect [19]. In the present study, a 13.2% (68.4%) higher rate of mothers compared to this study conducted in the USA stated that COVID-19 virus vaccination is safe for breastfeeding mothers.

Consistent with previous studies [20,21], in the present study, the vaccination rate was lower in families with low monthly household income. Again, most studies [18,22] determined that the vaccination rate was higher in those with higher education levels. A study in the USA stated that the low education level of black women did not change the vaccination rate. However, those with low education levels in white women were less vaccinated [23].

In the present study, an inverse correlation was found between the level of education and the belief that vaccinated people should stop breastfeeding. The rate (53.2%) of mothers who graduated from secondary school or below was significantly higher than that of mothers who graduated from high school or higher (30.2%).

A study conducted in Turkey determined vaccine hesitancy was higher in those who used the internet and social media intensively [24]. However, unlike this study, in the present study, the vaccination rate was higher in people who received information about the COVID-19 vaccine from the internet or their family physician. It may be because people think the pandemic process is well managed in Turkey. After all, most people prefer to follow the information about the pandemic from the official website of the Turkish Ministry of Health or reliable news sites. As a result, most people may have reached the correct information about the vaccine by following the internet and asking their family physician.

Contrary to studies showing that the acceptance of the COVID-19 vaccine increases as the number of children in the household increases, such a relationship was not found in the present study [18,19,25]. However, in the present study, there was a significantly higher rate of vaccination (85.7%) in those with children aged 0–6 months compared to those with children aged 7–24 months. It may be because COVID-19 virus vaccine was not initially recommended to pregnant women and breastfeeding mothers by the Turkish Ministry of Health. However, when the vaccine was found to be safe after widespread use and various studies, it was recommended to pregnant women in the period after the first trimester and breastfeeding mothers. While vaccine compatibility was high in young women in some studies [10,22,26], no relationship was found in the present study, as in the study of Levy et al. [23].

While it has been stated in the literature that people with COVID-19 virus infection in their family members and friends are more eager to get vaccinated due to fear of infection [17], such a relationship was not found in the present study. In addition, the vaccination rate (73.3%) in those with COVID-19 was significantly lower than in those without (86.3%).

In the present study, the rate of thinking that the baby should quit breast milk to be vaccinated against COVID-19 was significantly higher in mothers who were not vaccinated against COVID-19 (59.2%) compared to mothers who were vaccinated (40.1%). In addition, the rate of mothers who had a COVID-19 virus infection thinking that the baby should stop breastfeeding to be vaccinated against the COVID-19 virus (51.7%) was significantly higher than the rate of mothers who did not have a COVID-19 virus infection (40%). This result suggests that although 68.4% of mothers in the present study stated that it was safe for breastfeeding mothers to have the COVID-19 virus vaccine, many mothers did not get vaccinated in order not to stop breastfeeding. As a result, COVID-19 virus infection was more common in this group.

## 5. Conclusions

As with many diseases, breastfeeding women have a high risk of severe infection with the new COVID-19 virus. For this reason, it is vital for breastfeeding mothers to have their vaccinations, which is the best method of protection from the COVID-19 virus. The COVID-19 virus vaccine, which has been determined to be safe for pregnant and lactating women, should be administered without interruption or interruption of breast milk, a unique food in terms of maternal and child health.

The present study has shown that to eliminate the hesitancy about vaccination in mothers, it is necessary to inform and educate the whole society correctly, starting with families with low education and economic levels. Since the vaccination rate is higher in those who receive information from family doctor, all health professionals, especially family physicians, should spend more time informing them about vaccination. Contrary to the notion that the Internet increases anti-vaccination, our study has shown that the Internet helps to increase the vaccination rate, thanks to the sites that provide accurate information.

## Figures and Tables

**Table 1 vaccines-11-00813-t001:** Descriptive Characteristics of the Mothers Participating in the Study.

		Number	%
Maternal age	18–24 age	96	23.7
	25–34 age	282	69.6
	35 years of age and above	27	6.7
Educational status	Middle school and below	233	57.5
	High school and above	172	42.5
Economic situation	Good	128	31.6
	Medium	190	46.9
	Bad	87	21.5
Number of children	1	78	19.3
	2	165	40.7
	3 and above	162	40.0
Age of the child breastfed	0–6 month	265	65.4
	7–24 month	140	34.6
The status of having a new type of COVID-19 virus infection	Yes	120	29.6
	No	285	70.4
The status of having a new type of COVID-19 virus infection in their relatives	Yes	347	85.7
	No	58	14.3
Status of being vaccinated against COVID-19 virus	Yes	334	82.5
	No	71	17.5
COVID-19 virus vaccine information resource	Family practitioner	178	44.0
	Radio/TV	71	17.5
	Internet	86	21.2
	People around	70	17.3

**Table 2 vaccines-11-00813-t002:** Knowledge and Attitudes of 405 Mothers Participating in the Study About COVID-19 Virus Vaccination During Breastfeeding.

	Agree (%)	Disagree (%)	Hesitant (%)
Getting the COVID-19 virus vaccine is safe for breastfeeding mothers	68.4	17.8	13.8
I breastfed my baby 3 h after getting the COVID-19 virus vaccine	22.7	55.3	22.0
Breastfeeding mothers should be vaccinated	39.5	39.5	21.0
It does not upset me that I cannot get the COVID-19 virus vaccine because I am breastfeeding a baby.	33.3	46.9	19.8
COVID-19 virus vaccination of the breastfeeding mother also protects her baby	33.3	33.3	33.3
To be vaccinated against COVID-19 virus, the baby must be expected to quit breast milk.	38.8	42.0	19.3
I cannot get clear information from healthcare professionals about the vaccination of breastfeeding mothers.	31.9	33.3	34.8
I cannot get clear information from my family doctor about the vaccination of breastfeeding mothers.	44.0	34.8	21.2

**Table 3 vaccines-11-00813-t003:** COVID-19 virus vaccination status of the mothers participating in the study according to some descriptive characteristics.

		Vaccinated		Not Vaccinated		*p* *
		Sayı	%	Sayı	%	
Maternal age	18–24 age	77	80.2	19	19.8	0.778
	25–34 age	235	83.3	47	16.7	
	35 years of age and plus	22	81.5	5	18.5	
Educational status	Middle school and below	181	77.7	52	22.3	**0.003**
	High school and above	153	89.0	19	11.0	
Economical situation	Good	115	89.8 ^a^	13	10.2	**0.008**
	Medium	155	81.6 ^b^	35	18.4	
	Bad	64	73.6 ^c^	23	26.4	
Number of children	1	66	84.6	12	15.4	0.210
	2	141	85.5	24	14.5	
	3 and above	127	78.4	35	21.6	
Age of the child breastfed	0–6 month	227	85.7	38	14.3	**0.020**
	7–24 month	107	76.4	33	23.6	
The status of having a new type of COVID-19 virus infection	Yes	88	73.3	32	26.7	**0.002**
	No	246	86.3	39	13.7	
The status of having a new type of COVID-19 virus infection in their relatives	Yes	291	83.9	56	16.1	0.071
	No	43	74.1	15	25.9	
COVID-19 virus vaccine information resource	Family practitioner	156	87.6 ^a^	22	12.4	**0.035**
	Radio/TV	53	74.6 ^b^	18	25.4	
	Internet	72	83.7 ^a^	14	16.3	
	People around	53	75.7 ^b^	17	24.3	

* Chi-square analysis was applied. a,b,c. The group from which the difference originates. The bold mean significant *p* values.

**Table 4 vaccines-11-00813-t004:** According to the descriptive characteristics of the mothers participating in the study, the proportions of mothers who think that their baby should stop breastfeeding to be vaccinated against COVID-19.

		To Get the COVID-19 Virus Vaccine, the Baby Must Stop Breastfeeding.				
		Agree		Disagree-Hesitant		*p* *
		Number	%	Number	%	
Maternal age	18–24 age	44	45.8	52	54.2	0.725
	25–34 age	119	42.2	163	57.8	
	35 years of age and plus	13	48.1	14	51.9	
Education	Middle school and below	124	53.2	109	46.8	**<0.001**
	High school and above	52	30.2	120	69.8	
Economical situation	Good	58	45.3	70	54.7	0.500
	Medium	85	44.7	105	55.3	
	Bad	33	37.9	54	62.1	
Number of children	1	29	37.2	49	62.8	0.461
	2	74	44.8	91	55.2	
	3 and plus	73	45.1	89	54.9	
Age of the child breastfed	0–6 month	109	41.1	156	58.9	0.194
	7–24 month	67	47.9	73	52.1	
The status of having a new type of COVID-19 virus infection	Yes	62	51.7	58	48.3	**0.031**
	No	114	40.0	171	60.0	
The status of having a new type of COVID-19 virus infection in their relatives	Yes	147	42.4	200	57.6	0.277
	No	29	50.0	29	50.0	
Status of being vaccinated against COVID-19 virus	Yes	134	40.1	200	59.9	**0.003**
	No	42	59.2	29	40.8	
COVID-19 virus vaccine information resource	Family practitioner	74	41.6	104	58.4	0.216
	Radio/TV	37	52.1	34	47.9	
	Internet	40	46.5	6	53.5	
	People around	25	35.7	45	64.3	

* Chi-square analysis was applied. The bold mean significant *p* values.

**Table 5 vaccines-11-00813-t005:** Comparison of information sources of various parameters.

		Family Doctor		Radio/TV		Internet		People Around		*p* *
		n	%	n	%	n	%	n	%	
Maternal age	18–24 age	50	52.1	18	18.8	19	19.8	9	9.4	0.109
	25–34 age	114	40.4	48	17.0	65	23.0	55	19.5	
	35 and above	14	51.9	5	18.5	2	7.4	6	22.2	
Educational status	Middle school and below	80	34.3 ^a^	51	21.9 ^a^	56	24.0 ^a^	46	19.7 ^a^	**<0.001**
	High school and above	98	57.0 ^b^	20	11.6 ^b^	30	17.4 ^a^	24	14.0 ^a^	
Economical situation	Good	58	45.3	26	20.3	26	20.3	18	14.1	0.887
	Moderate	84	44.2	30	15.8	41	21.6	35	18.4	
	Bad	36	41.4	15	17.2	19	21.8	17	19.5	
COVID-19 status	Yes	78	65.0 ^a^	14	11.7 ^a^	19	15.8 ^a^	9	7.5 ^a^	**<0.001**
	No	100	35.1 ^b^	57	20.0 ^b^	67	23.5 ^a^	61	21.4 ^b^	

* Chi-square analysis was applied. ^a,b^ Source of differences. The bold mean significant *p* values.

## Data Availability

Data will be available upon request from the corresponding author.

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
