# Peer review of "COVID-19 Vaccination and Breastfeeding Mothers in Kahta District, Turkey"

_vaccines, 2023, doi:10.3390/vaccines11040813_

Round 1

Reviewer 1 Report

I was pleased to read the Manuscript by Parlak et al. on "Knowledge, Attitudes, and Behaviors of Breastfeeding Mothers About COVID-19 Vaccination" and I think it covers a topic of great current interest to the scientific community. Below are my comments:

Title

I recommend reviewing it and avoiding overstatement. It might be appropriate to include the study design. Definitely needed to circumscribe it e.g. adding "in Katha district, Turkey."

Abstract

L17, L19, etc. when writing "significantly higher" it is good to report epidemiological measures to support the statement. CI95%? p-value?

L24-26 I do not think these considerations adhere to the results of the study. I think it would be more useful to summarize along these lines what the utility of this study is to the scientific community/clinicians.

Background

I suggest making the introduction more exhaustive, at the moment it has only enough background information.

Methods

Need to be made more complete and clearer. Were all women entering the clinic enrolled? If so, is the proportion of those with high education and those with low education reflecting the general population?

I think it is necessary to better explain the recruitment methods and also to include all these considerations about the non-representativeness of the sample in the limitations section. The limitations section is currently absent; it is necessary to write a paragraph in which all the limitations of the study are thickened one by one. 

Also, what is the non-acceptance rate of the study?

Conclusions

I would stick more to the results and avoid considerations that do not emerge from this study. The fact that breastfeeding is important is certainly an important element and well put in the introduction and discussion. In this section, it is better to focus on the main findings, utility, and learnings of this particular study. 

Author Response

RESPONSES TO THE REVIEWER-1

Comments and Suggestions for Authors

I was pleased to read the Manuscript by Parlak et al. on "Knowledge, Attitudes, and Behaviors of Breastfeeding Mothers About COVID-19 Vaccination" and I think it covers a topic of great current interest to the scientific community. Below are my comments:

  • Dear reviewer, we would like to thank you for your valuable comments and suggestions to our study. We believe that the scientific quality of the text which was completed in accordance with your suggestions increased.

Title

I recommend reviewing it and avoiding overstatement. It might be appropriate to include the study design. Definitely needed to circumscribe it e.g. adding "in Katha district, Turkey."

  • The title was changed in accordance with your suggestion.

Abstract

L17, L19, etc. when writing "significantly higher" it is good to report epidemiological measures to support the statement. CI95%? p-value?

  • The significance level of the finding was added to the text using with p-value.

L24-26 I do not think these considerations adhere to the results of the study. I think it would be more useful to summarize along these lines what the utility of this study is to the scientific community/clinicians.

  • The section was re-phrased in accordance with your comments.

Background

I suggest making the introduction more exhaustive, at the moment it has only enough background information.

  • Thank you for valuable comments. We added some additional details in the introduction of the text.

Methods

Need to be made more complete and clearer. Were all women entering the clinic enrolled? If so, is the proportion of those with high education and those with low education reflecting the general population?

  • Dear reviewer, you are totally right. We also think that the sample education levels should reflect the population. We explained the sample of the study using more clear informations in the method section.

I think it is necessary to better explain the recruitment methods and also to include all these considerations about the non-representativeness of the sample in the limitations section. The limitations section is currently absent; it is necessary to write a paragraph in which all the limitations of the study are thickened one by one. 

  • We explained the limitations of the study at the end of methods section.

Also, what is the non-acceptance rate of the study?

  • The non-acceptance rate of the study is 3.8%, and the information is added to the methods section.

Conclusions

I would stick more to the results and avoid considerations that do not emerge from this study. The fact that breastfeeding is important is certainly an important element and well put in the introduction and discussion. In this section, it is better to focus on the main findings, utility, and learnings of this particular study. 

  • Dear reviewer, We totally agree with you. We rephrased some parts of the discussion section, Some additional critics were added the discussion section of the text.

Reviewer 2 Report

Line 16-17 has an error: “The vaccination rate (85.7%) of children whose breastfed child was 0-6 months old was found to be significantly higher than that of 7-24 months old (76.4%)”. It is not expected that children have breastfed children.

General language suggestion: change coronavirus / vaccine to SARS-CoV-2 or Covid virus / vaccine, since at least 6 other human coronaviruses exist.

Table 2 should include the number of participants.

It would be good to inform the reader on practices in Turkey concerning vaccination during pregnancy – is it recommended or not - Did it change recently, which could explain the difference between having a 0-6 and 7-24 months old baby e.g. ~ 6-15 months before the study. This would be invaluable background information and provides input for a valuable discussion.

It would be nice if interactions between parameters would also be examined e.g., is educational level associated with internet use?

Discussion might be improved by some minor speculation on causes, specifically for Turkey e.g., why internet seems to be reliable source.

Author Response

RESPONSES TO THE REVIEWER-2

  • Dear reviewer, we would like to thank you for your valuable comments and suggestions to our study. We believe that the scientific quality of the text which was completed in accordance with your suggestions increased.

Comments and Suggestions for Authors

Line 16-17 has an error: “The vaccination rate (85.7%) of children whose breastfed child was 0-6 months old was found to be significantly higher than that of 7-24 months old (76.4%)”. It is not expected that children have breastfed children.

  • Dear reviewer, we apologize for this misspelling that we overlooked. The sentence was corrected.

General language suggestion: change coronavirus / vaccine to SARS-CoV-2 or Covid virus / vaccine, since at least 6 other human coronaviruses exist.

  • We used only one form of expression in the revised text.

Table 2 should include the number of participants.

  • The number of participants was added in Table-2.

It would be good to inform the reader on practices in Turkey concerning vaccination during pregnancy – is it recommended or not - Did it change recently, which could explain the difference between having a 0-6 and 7-24 months old baby e.g. ~ 6-15 months before the study. This would be invaluable background information and provides input for a valuable discussion.

  • Thank you for your valuable comment. We added some additional critics in the discussion section.

It would be nice if interactions between parameters would also be examined e.g., is educational level associated with internet use?   

  • Dear reviewer, you are totally right. Like you, we thought that there might be a relationship between some sociodemographic variables such as education level and internet use. However, in the initial analysis of the data, we had to include only the statistically significant findings, keeping in mind some important rules of the special issue.

Discussion might be improved by some minor speculation on causes, specifically for Turkey e.g., why internet seems to be reliable source.

  • We rephrased some parts of the discussion section. All revisions are marked with colorful characters. What is more, all revisions to the manuscript are marked up using the “Track Changes” function.

Round 2

Reviewer 1 Report

I thank the author for having addressed all my comment point by point.